# Perceived change in physical activity levels and mental health during COVID-19: Findings among adult twin pairs

**Glen E. Duncan**[1]*, **Ally R. Avery**[1], **Edmund Seto**[2], **Siny Tsang**[1]

**1** Department of Nutrition and Exercise Physiology, Washington State University Health Sciences Spokane, Spokane, Washington, United States of America, **2** Department of Environmental and Occupational Health Sciences, University of Washington, Seattle, Washington, United States of America

* glen.duncan@wsu.edu

**Data Availability Statement:** The data supporting the results of the present study are owned by the Washington State Twin Registry (WSTR). Thus, the data cannot be publicly shared as it involves third-

## Abstract

### Background

Physical distancing and other COVID-19 pandemic mitigation strategies may have unintended consequences on a number of health behaviors and health outcomes. The purpose of this study was to investigate the association between perceived change in physical activity or exercise and mental health outcomes over the short-term in response to COVID-19 mitigation strategies in a sample of adult twins.

### Methods

This was a cross-sectional study of 3,971 identical and same-sex fraternal adult twins (909 pairs, 77% identical) from the community-based Washington State Twin Registry. Participants in this study completed an online survey examining the impact of COVID-19 mitigation on a number of health-related behaviors and outcomes, administered between March 26 and April 5, 2020. In the present study, the exposure was perceived change in physical activity or exercise. The outcomes were levels of perceived anxiety and stress.

### Results

More twin pairs reported a decrease in physical activity levels (42%) than those reporting no change (31%) or increased physical activity levels (27%). A perceived decrease in physical activity or exercise was associated with higher stress and anxiety levels. However, the physical activity–stress relationship was confounded by genetic and shared environmental factors. On the other hand, the physical activity–anxiety relationship held after controlling for genetic and shared environmental factors, although it was no longer significant after further controlling for age and sex, with older twins more likely to report lower levels of anxiety and females more likely to report higher levels of anxiety.

### Conclusions

Strategies to mitigate the COVID-19 pandemic may be impacting physical activity and mental health, with those experiencing a decrease in physical activity also having higher levels

party data. However, researchers interested to apply to gain access to the data can do so by contacting the WSTR and completing the appropriate forms stipulated in the WSTR Policies & Procedures guidelines. Application information can be sent to the Scientific Operations Manager at the following URL (https://wstwinregistry.org/contact-us/) or via email (ws.twinregistry@wsu.edu).

**Funding:** ES is supported by NIH grants R33 ES024715-04 and 5P30 ES007033-23, and GED is supported by NIH grant R33 ES024715-04.

**Competing interests:** The authors have declared that no competing interests exist.

of stress and anxiety. These relationships are confounded by genetic and shared environmental factors, in the case of stress, and age and sex, in the case of anxiety.

## Introduction

The novel coronavirus (SARS-CoV-2 virus and associated disease COVID-19, abbreviated as COVID-19 throughout) has disrupted the daily lives of people around the globe. In the United States (U.S.), physical distancing and related closures of businesses and activity spaces (e.g., public and private gymnasiums, athletic fields) have been widely instituted to mitigate the spread of COVID-19. However, these strategies may have unintended consequences on health behaviors and health outcomes.

Regular physical activity is a cornerstone of chronic disease prevention and treatment. Engaging in an active lifestyle is associated with a number of important mental health outcomes [1, 2]. In contrast, physical inactivity is associated with poor mental health outcomes [3–5]. These associations are generally robust after controlling for variables that are known to confound the activity–mental health relationship.

There is global concern for the impacts of COVID-19 quarantine and physical distancing measures on health behaviors and physical and mental health [6–12], although few studies have addressed these concerns empirically. However, a few recent reports indicate there are impacts of COVID-19 mitigation strategies on psychological symptoms [13] and physical activity levels [14–17]. One study, conducted in 1,491 Australian adults, reported that negative changes in physical activity, sleep, smoking, and alcohol were associated with higher depression, anxiety, and stress symptoms [18]. Building on this prior evidence, the purpose of the present study was to investigate the association between perceived change in the amount of physical activity or exercise and mental health over the short-term in response to COVID-19 mitigation strategies in the U.S. We hypothesized that a decrease in the amount of physical activity or exercise would be associated with higher levels of stress and anxiety. Conversely, we hypothesized that an increase in the amount of physical activity or exercise would be associated with lower levels of stress and anxiety. The current study investigated perceived changes in physical activity and mental health in response to COVID-19 mitigation strategies in identical (MZ) and fraternal (DZ) adult twins. This is an important consideration because the perceived change in physical activity and its association with current mental health, within the context of COVID-19 restrictions, are expected to be primarily attributable to environmental factors that are unique, or non-shared, between individuals. Using twins allowed us to control for genetic and shared environmental factors that are associated with both perceived changes in physical activity levels and mental health. We were therefore able to examine whether associations between perceived changes in physical activity and mental health were confounded by genetic and shared environmental factors [19].

## Materials and methods

### Participants

The current study included 3,971 individuals from the Washington State Twin Registry (WSTR). Details regarding the WSTR are reported elsewhere [20–22]. Participants in this study completed an online survey examining a number of health-related behaviors and outcomes and their impact due to COVID-19 mitigation, administered between March 26 and April 5, 2020. The study was approved by the IRB at Washington State University. A wavier of

documentation of consent was obtained from the IRB, and consent was thus assumed by completing the questionnaire. The survey was sent to 12,173 twins; the individual response rate was 32.8% and the pair-wise response rate was 21.2%. These response rates were comparable to prior WSTR survey-based studies (about 32% and 21% individual and pair-wise response rates, respectively, across 13 studies). Demographic characteristics of sample respondents were similar to those in the full WSTR.

Among those who participated in the survey, there were 909 same-sex twin pairs (77% MZ, 23% DZ). Zygosity was determined using five questions in the WSTR enrollment survey asking about childhood similarity. Compared to biological zygosity indicators, the survey items correctly classify zygosity with at least 95% accuracy [23, 24].

## Measures

The independent variable was perceived change in physical activity or exercise, assessed by asking participants the following question: "Compared to one month ago (i.e., prior to the spread of COVID-19), how much has your daily life changed in the following areas?". A number of behaviors and activities were assessed; for the present study, we focused on responses to "Amount of physical activity or exercise", with five possible response categories (*decreased a lot*, *decreased somewhat*, *no change*, *increased somewhat*, and *increased a lot*).

Considering the small proportions of participants who responded their amount of physical activity *decreased a lot* (15.1% and 14.0% in full and same-sex twins sample, respectively) or *increased a lot* (5.2% and 5.4% in full and same-sex twins sample, respectively), these two categories were respectively combined with *decreased somewhat* and *increased somewhat*. Participants were thus categorized into three physical activity change groups: decreased, no change, and increased.

Mental health outcomes included stress and anxiety. Stress was assessed using the 10-item Perceived Stress Scale (PSS) [25]. Participants were asked about their feelings and thoughts in the last two weeks including the day the participant completed the survey on a 5-point Likert-type scale (0 = *Never*; 1 = *Almost never*; 2 = *Sometimes*; 3 = *Fairly often*; 4 = *Very often*). After reverse-coding responses to the four positively stated items, PSS scores were obtained by summing across all scale items, with higher scores reflecting higher levels of perceived stress (range = 0 to 40). Reliability for this scale was very good (Chronbach's alpha = 0.89, 95% CI = 0.88–0.90).

Anxiety was assessed using the six-item anxiety subscale in the Brief Symptom Inventory (BSI) [26]. Participants were asked to indicate how much discomfort each problem has caused them during the past two weeks including the day the participant completed the survey on a 5-point Likert-type scale (0 = *Not at all*; 1 = *A little bit*; 2 = *Moderately*; 3 = *Quite a bit*; 4 = *Extremely*). Anxiety scores were obtained by summing across all items, with higher scores reflecting higher levels of anxiety (range = 0 to 24). Reliability for this scale was also very good (Chronbach's alpha = 0.88, 95% CI = 0.87–0.88).

Participants' age and sex were included as covariates. Age was computed based on the reported date of birth and date of survey completion. Sex was self-reported as man or woman.

## Statistical analysis

Considering the non-linear associations between perceived change in physical activity or exercise and perceived stress and anxiety (S1 Fig in S1 File), we determined that separate comparisons should be made between those who reported a change and those who reported no change in the independent variable. Two comparisons were performed for each outcome: decreased versus no change and increased versus no change, for a total of four comparisons.

We first used the classical twin model to decompose the variances of change in the amount of physical activity, perceived stress, and anxiety into additive genetic (A), shared environmental (C), and non-shared environmental (E) components [27]. Next, we used phenotypic regression models (labeled Model 1 in the data tables) to examine the association between change in physical activity and mental health outcomes (S2 Fig in S1 File). Perceived stress and anxiety were separately regressed on change in physical activity to estimate the observed association between the independent variable and each mental health outcome (labeled $b_{phen}$ in S2 Fig in S1 File, reflecting the phenotypic or observed association between change in physical activity and mental health, without including genetic or shared environmental confounds).

The models were then re-estimated including estimates of $b_A$ and/or $b_C$, which controls for genetic and shared environmental confounds, respectively, in the estimation of the phenotypic association (S2 Fig in **S1 File**). These are referred to as "quasi-causal" models (labeled Model 2); the logic and associated statistical methods are described in detail elsewhere [19]. A final set of models were estimated by including participants' age and sex as covariates (labeled Model 3). Perceived stress and anxiety were both square root transformed because they were positively skewed. Age was divided by 10 to allow variables to be on similar scales.

Descriptive statistics were computed for both the full sample and the same-sex twin sample, whereas twin analyses were performed only on the same-sex twin sample, using R 3.5.3 (R Core Team, 2013). All latent variable path analyses were conducted using the computer program Mplus v8.1. The alpha level for testing hypotheses was set to 0.05 *a priori*. Twin-based regression models are generally saturated; the only source of reduced fit involves incidental issues such as differences between twins arbitrarily assigned as Twin 1 and Twin 2 within pairs. All reported models fit the data closely using standard "goodness of fit" tests.

## Results

Descriptive statistics for select demographic characteristics, perceived change in the amount of physical activity or exercise, perceived stress, and anxiety are reported for the full sample and for same-sex twin pairs (Table 1). In S1 Text and S1 Table in **S1 File**, we provide information on the twin correlations for the exposure and outcome variables.

### Change in physical activity and perceived stress

**Increase vs. no change.** We found no evidence of an association between perceived change in physical activity or exercise and perceived stress when comparing twins who reported an increase and those who reported no change in physical activity (Table 2). Results were similar in both the phenotypic model (Model 1: $b_{phen}$ = 0.089, standard error (SE) = 0.060, p = 0.139), the quasi-causal model (Model 2: $b_{phen}$ = 0.024, SE = 0.044, p = 0.577), and the quasi-causal model controlling for age and sex (Model 3: $b_{phen}$ = 0.016, SE = 0.028, p = 0.568).

**Decrease vs. no change.** Results from the phenotypic model (Model 1 in Table 2) showed a significant association between change in physical activity and perceived stress when comparing individuals who reported a decrease and those who reported no change in physical activity ($b_{phen}$ = 0.036, SE = 0.010, p < 0.001). When between-family confounds were controlled in the quasi-causal model (Model 2), the association was reduced and became non-significant ($b_{phen}$ = 0.017, SE = 0.010, p = 0.090), suggesting that genetic and shared environmental factors confounded the association between a perceived decrease in physical activity and perceived stress. Results remained similar after controlling for age and sex (Model 3: $b_{phen}$ = 0.050, SE = 0.033, p = 0.132).

The phenotypic association between a decrease in physical activity and perceived stress is illustrated in Fig 1. Concordant decreasing twin pairs (i.e., both members of the pair reported

**Table 1. Descriptive statistics of select demographic characteristics, perceived change in physical activity or exercise, perceived stress, and anxiety, among twins in the Washington State Twin Registry.**

|  | Full sample (*N* = 3,971) | Same-sex twin pairs (*n* = 909 pairs) |
|---|---|---|
| **Age** | 50.4 (16.0) | 49.9 (16.0) |
| **Sex** | | |
| **Men** | 1,125 (30.8%) | 444 (24.4%) |
| **Women** | 2,746 (69.2%) | 1,374 (75.6%) |
| **White (non-Hispanic)** | 3,793 (95.5%) | 1,738 (95.6%) |
| **Zygosity** | | |
| **MZ** | 2,385 (60.1%) | 1,400 (77.0%) |
| **DZ** | 1,586 (39.9%) | 418 (23.0%) |
| **Change in physical activity** | | |
| **Decreased** [a] | 1,735 (43.8%) | 771 (42.4%) |
| **No change** | 1,045 (26.4%) | 557 (30.6%) |
| **Increased** [b] | 1,183 (29.8%) | 490 (27.0%) |
| **Perceived stress** | 12.3 (7.2) | 12.6 (7.2) |
| **Anxiety** | 3.6 (3.6) | 3.8 (4.0) |

Means (standard deviations) are presented for continuous variables. Frequencies (proportions) are presented for categorical variables.

[a]In the full sample, 15.1% reported "decreased a lot" and 28.7% reported "decreased somewhat." In the same-sex twin sample 14.0% reported "decreased a lot" and 28.4% reported "decreased somewhat.

[b]In the full sample, 5.2% reported "increased a lot" and 21.2% reported "increased somewhat." In the same-sex twin sample, 5.4% reported "increased a lot" and 21.6% reported "increased somewhat."

a decrease in physical activity) showed higher levels of perceived stress (leftmost bar in each panel) than concordant no change twin pairs (i.e., both members of the pair reported no change in physical activity; rightmost bar in each panel). The pattern was consistent between MZ (left panel) and DZ pairs (right panel). However, within pairs discordant for change in physical activity (middle two bars in each panel), there was little visible difference in perceived stress levels between members of the pair who reported a decrease (second bar from the left) and co-twins who reported no change in physical activity (third bar from the left).

### Change in physical activity and anxiety

**Increase vs. no change.** We found no evidence of an association between perceived change in physical activity and anxiety when comparing twins who reported an increase and those who reported no change in physical activity (Table 3). Results were similar in the phenotypic model (Model 1: $b_{phen}$ = 0.117, *SE* = 0.079, p = 0.141), the quasi-causal model (Model 2: $b_{phen}$ = 0.085, *SE* = 0.068, p = 0.212), and the quasi-causal model controlling for age and sex (Model 3: $b_{phen}$ = 0.047, *SE* = 0.040, p = 0.236).

**Decrease vs. no change.** There was a significant association between a decrease in physical activity and anxiety (Model 1: $b_{phen}$ = 0.143, *SE* = 0.039, p < 0.001); individuals who reported a decrease in physical activity showed higher levels of anxiety than those who reported no change (Table 3). The association remained significant after controlling for between-family confounds ($b_{phen}$ = 0.134, *SE* = 0.042, p = 0.002). As shown in Fig 2, concordant twin pairs who reported a decrease in physical activity had higher levels of anxiety (leftmost bar in each panel) than concordant twin pairs who reported no change in physical activity (rightmost bar in each panel). This association was also evident within MZ pairs

**Table 2. Unstandardized parameter estimates for phenotypic and biometric models estimating associations between self-reported change in physical activity or exercise and perceived stress.**

| | | Model 1 | | Model 2 | | Model 3 | |
|---|---|---|---|---|---|---|---|
| | | Phenotypic model | | Quasi-causal model | | Quasi-causal model | |
| | | Est (SE) | p | Est (SE) | p | Est (SE) | p |
| **Increase vs. no change** | | | | | | | |
| | $b_A$ | | | .152 (.115) | .118 | .044 (.049) | .374 |
| | $b_{phen}$ | .089 (.060) | .139 | .024 (.044) | .577 | .016 (.028) | .568 |
| | Age | | | | | -.272 (.017) | < .001 |
| | Sex (F) | | | | | .448 (.070) | < .001 |
| **RMSEA [90%CI]** | | .020 [0, .049] | | .018 [0, .049] | | .032 [.009, .054] | |
| **Decrease vs. no change** | | | | | | | |
| | $b_C$ | | | .074 (.030) | .015 | .166 (.084) | .049 |
| | $b_{phen}$ | .036 (.010) | < .001 | .017 (.010) | .090 | .050 (.033) | .132 |
| | Age | | | | | -.271 (.017) | < .001 |
| | Sex (F) | | | | | .447 (.070) | < .001 |
| **RMSEA [90%CI]** | | .020 [0, .049] | | .013 [0, .46] | | .036 [.016, .053] | |

Standard errors are presented within parentheses. Phenotypic model does not include controls for between-pair confounds, whereas quasi-causal model include controls for between-pair confounds. Perceived stress is square root transformed; age is divided by 10.

$b_A$: amount of variance in perceived stress attributable to additive genetic influences; $b_C$: amount of variance in perceived stress attributable to shared environmental influences; $b_{phen}$: phenotypic association between predictor and outcome; *RMSEA*: root mean square error of approximation.

discordant for change in physical activity. Members of the twin pair who reported decreased physical activity had higher levels of anxiety (second bar from the left in the left panel) than co-twins who reported no change in physical activity (third bar from the left in the left panel),

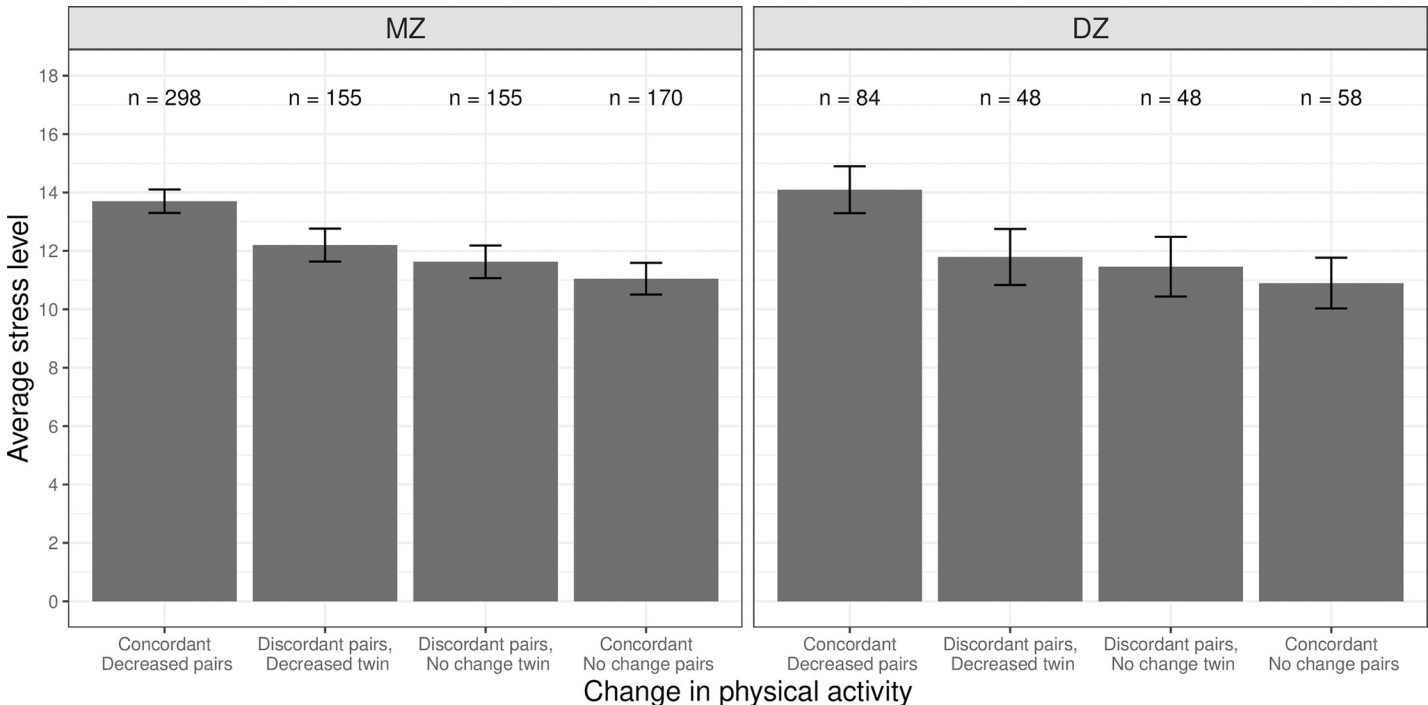

**Fig 1. Average perceived stress levels between twin pairs concordant and discordant in change in physical activity or exercise among same-sex twin pairs.**
MZ = monozygotic (identical); DZ = dizygotic (fraternal), Error bars denote standard errors.

**Table 3. Unstandardized parameter estimates for phenotypic and biometric models estimating associations between self-reported change in physical activity or exercise and anxiety.**

| | | Model 1 | | Model 2 | | Model 3 | |
|---|---|---|---|---|---|---|---|
| | | Est (*SE*) | *p* | Est (*SE*) | *p* | Est (*SE*) | *p* |
| **Increase vs. no change** | | | | | | | |
| | $b_A$ | | | .146 (*.119*) | .218 | .039 (*.047*) | .396 |
| | $b_{phen}$ | .117 (*.079*) | .141 | .085 (*.068*) | .212 | .047 (*.040*) | .236 |
| | Age | | | | | -.221 (*.018*) | < .001 |
| | Sex (F) | | | | | .612 (*.069*) | < .001 |
| **RMSEA [90%CI]** | | .022 [0, .050] | | .024 [0, .052] | | .034 [.012, .052] | |
| **Decrease vs. no change** | | | | | | | |
| | $b_C$ | | | .149 (*.086*) | .064 | .133 (*.114*) | .245 |
| | $b_{phen}$ | .143 (*.039*) | < .001 | .134 (*.042*) | .002 | .150 (*.106*) | .158 |
| | Age | | | | | -.218 (*.017*) | < .001 |
| | Sex (F) | | | | | .613 (*.069*) | < .001 |
| **RMSEA [90%CI]** | | .030 [0, .056] | | .032 [0, .058] | | .044 [.027, .061] | |

Standard errors are presented within parentheses. Phenotypic model does not include controls for between-pair confounds, whereas quasi-causal model include controls for between-pair confounds. Perceived stress is square root transformed; age is divided by 10.

$b_A$: amount of variance in perceived stress attributable to additive genetic influences; $b_C$: amount of variance in perceived stress attributable to shared environmental influences; $b_{phen}$: phenotypic association between predictor and outcome; *RMSEA*: root mean square error of approximation.

suggesting a quasi-causal association between decrease in physical activity and anxiety. However, this association was attenuated and no longer significant upon controlling for age and sex (Model 3: $b_{phen}$ = 0.150, *SE* = 0.106, p = 0.158).

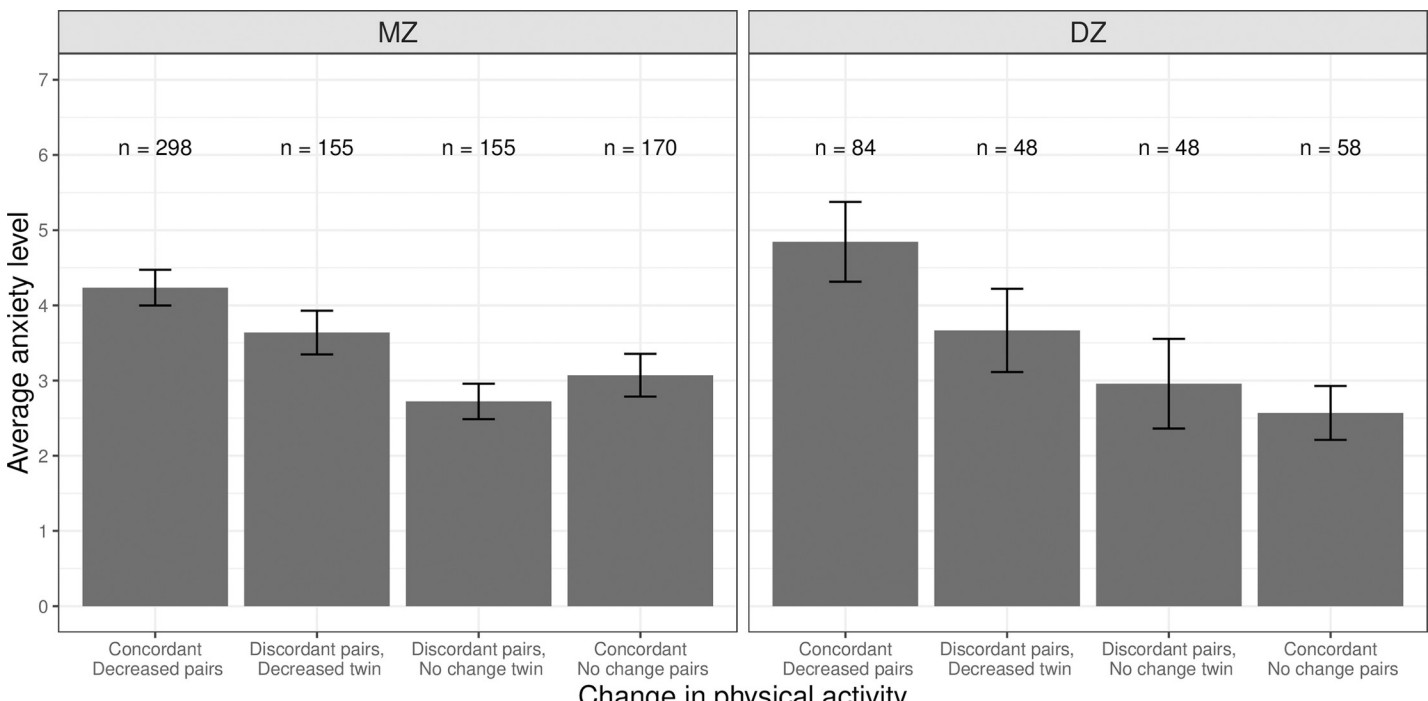

**Fig 2. Average anxiety levels between twin pairs concordant and discordant in change in physical activity or exercise among same-sex twin pairs.**
MZ = monozygotic (identical); DZ = dizygotic (fraternal), Error bars denote standard errors.

## Discussion

Results of the present study support the first hypothesis that a decrease in the perceived amount of physical activity or exercise was associated with higher levels of perceived stress and anxiety. Individuals who reported a decrease in physical activity, in the context of COVID-19 mitigation strategies, were compared to those who reported no change. The phenotypic (between-twin model) association between a decrease in physical activity and stress was attenuated and no longer significant after controlling for between family confounds ("quasi-causal", within-twin model), suggesting that genetic and shared environmental factors confound this relationship. On the other hand, the phenotypic association between a decrease in physical activity and anxiety remained significant after controlling for genetic and shared environmental factors. However, this association was no longer significant after further controlling for age and sex, suggesting that any within-family confounding on anxiety was small.

Older twins were more likely to report slightly lower levels of anxiety ($<$ 1-point difference in anxiety per 10-year increase in age), and females were more likely to report slightly higher levels of anxiety ($<$ 0.5 points higher) than males. Interestingly, the proportion of individuals reporting different changes in physical activity did not differ by sex (45.3% decrease, 29.8% no change, and 25.5% increase for males: 41.5% decrease, 31.1% no change, and 27.4% increase for females). Nonetheless, the small proportion of men in this study precludes us from investigating whether the association between change in physical activity and mental health differs by sex.

An increase in the perceived amount of physical activity or exercise was not associated with lower levels of stress and anxiety, providing no support for the second hypothesis. In fact, as shown in S1 Fig in **S1 File**, individuals who reported an increase in physical activity had slightly higher levels of stress (mean [denoted by $M$], $M_{Increase} = 12.8$ versus $M_{NoChange} = 11.4$) and anxiety ($M_{Increase} = 4.0$ versus $M_{NoChange} = 2.9$) than those who reported no change. The difference was not significant, possibly due to the large variance in stress ($SD = 7.1$ and $7.0$ for increase and no change groups, respectively) and anxiety ($SD = 4.1$ and $3.4$ for increase and no change groups, respectively) in both groups.

As noted in a recent *Viewpoint* (published in *JAMA Intern Med*), stay at home orders across the U.S. (and elsewhere) to mitigate the spread of COVID-19 will undoubtedly have consequences for mental health and well-being in both the short and longer term [28]. In this context, at least in the short-term, possible reasons that many individuals reported a decrease in physical activity include having a difficult time managing multiple duties, such as balancing work at home with child care, or lacking resources for physical activity, such as equipment at home or living in neighborhoods that do not support being physically active. On the other hand, these measures may also have been the reason why a small proportion of individuals reported an increase in physical activity. As most individuals are no longer commuting to work and/or participating in their daily routines, some may choose to engage in physical activity (e.g., bicycling with children) as an opportunity to leave their homes. Some may even find the additional physical activity, regardless of duration or intensity, to be a helpful coping strategy.

In order to better understand the long-term impact of COVID-19 mitigation strategies on health behaviors and outcomes, longitudinal studies investigating changes in daily activities and health are needed. As physical restrictions are lifted, it will also be important to examine whether physical activity and mental health levels return to pre-COVID-19 levels, and whether these changes differ among individuals with different baseline levels of physical activity, stress, and anxiety.

## Strengths and limitations

The primary strength of the present study is the timeliness of the survey. Data collection occurred between the last week of March and the first week of April 2020. Washington implemented the state-wide "stay home, stay healthy" order on March 24, 2020. The timing of the survey thus allowed us to capture the immediate impact of physical distancing measures and other mitigation strategies on perceived changes in physical activity and mental health in a relatively large sample.

Our use of twin pairs as participants is another strength because it allowed us to control for genetic and shared environmental factors that might otherwise confound the physical activity–mental health relationship. This level of control is not available in studying unrelated individuals, and thus our use of twin modeling provides robust evidence that any identified associations between exposures and outcomes of interest are "quasi-causal" rather than due to confounding and chance alone. Nonetheless, causal inference is still limited in the present study due to its observational design.

On the other hand, physical restriction measures in the U.S. varied greatly by state, and even by county, thus resulting in likely differences in how changes in activity levels were perceived. It was also difficult to identify specific timepoints for which the participants were to evaluate changes in the amount of physical activity for comparison purposes. In addition, at the time the survey was administered, most individuals were still adjusting to their new routines (e.g., working from home). It was impossible to assess the amount of physical activity they did *in a typical week*, as there was nothing *typical* once COVID-19 was declared a pandemic by the World Health Organization (WHO) on March 11, 2020.

As with any survey research, the current study potentially suffers from self-selection bias. Average stress and anxiety levels were relatively low; the average stress level was 12.6 ($SD = 7.2$) out of a maximum possible score of 40, and the average anxiety level was 3.8 ($SD = 4.0$) out of a maximum possible score of 24. It is therefore possible that individuals who chose to participate in this survey had lower levels of stress and anxiety overall, whereas those who were more stressed and anxious during this time opted not to participate. However, without information on non-responders, we are unable to speculate whether individuals with lower levels of stress and anxiety self-selected to participate in this study. It is also unknown whether different patterns of association would be found among those with extreme levels of stress and anxiety, thus limiting generalizability to samples with different demographic characteristics. Finally, as the current study relied on self-reported data, it is possible that our results may be affected by self-report bias.

## Conclusions

We found that a perceived decrease in physical activity or exercise was associated with increased perceived stress and anxiety levels, however, these associations were relatively small and confounded by between-family factors and demographic characteristics such as age and sex. Overall, our findings suggest that physical distancing mitigation strategies in response to the COVID-19 pandemic may have an impact on individuals' daily activities and mental health, specifically with slightly higher levels of stress and anxiety among those who experienced a decreased amount of physical activity. The WSTR is conducting follow-up studies to investigate the extent to which the amount of physical activity varies as physical restrictions are slowly lifted; these longitudinal studies will allow us to follow up on the present cross-sectional findings and further determine associations between changes in health behavior exposures and changes in mental health outcomes as the pandemic unfolds.

## Supporting information

**S1 File.**
(DOCX)

## Acknowledgments

We thank the twin members of the Washington State Twin Registry for their participation in our research.

## Author Contributions

**Conceptualization:** Glen E. Duncan.

**Data curation:** Ally R. Avery, Siny Tsang.

**Formal analysis:** Siny Tsang.

**Investigation:** Glen E. Duncan.

**Methodology:** Glen E. Duncan, Ally R. Avery, Edmund Seto, Siny Tsang.

**Writing – original draft:** Glen E. Duncan.

**Writing – review & editing:** Glen E. Duncan, Ally R. Avery, Edmund Seto, Siny Tsang.

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
