## [Decision Letter · Decision Letter 0]

17 Jul 2020

PONE-D-20-20044

Perceived change in physical activity levels and mental health during COVID-19: Findings among adult twin pairs

PLOS ONE

Dear Dr. Duncan,

Thank you for submitting your manuscript to PLOS ONE. After careful consideration, we feel that it has merit but does not fully meet PLOS ONE’s publication criteria as it currently stands. Therefore, we invite you to submit a revised version of the manuscript that addresses the points raised during the review process.

Please see Additional Editor Comments.

We look forward to receiving your revised manuscript.

Kind regards,

Michio Murakami

Academic Editor

PLOS ONE

Additional Editor Comments:

1. I recommend the authors to use "physical distancing" rather than "social distancing," if they have no special reasons.

2. The authors need to add a limitation that the causalty is questionable as a nature of observational study.

2. Please note that according to our data availability statement (https://journals.plos.org/plosone/s/data-availability), PLOS does not permit references to “data not shown.” or "data available upon request".

Please provide the relevant data within the manuscript, the Supporting Information files, or in a public repository. If the data are not a core part of the research study being presented, please remove any references to these data.

Thank you for your attention to this request.

Reviewers' comments:

Reviewer's Responses to Questions

**Comments to the Author**

1. Is the manuscript technically sound, and do the data support the conclusions?

Reviewer #1: Yes

Reviewer #2: Yes

2. Has the statistical analysis been performed appropriately and rigorously? 

Reviewer #1: I Don't Know

Reviewer #2: Yes

3. Have the authors made all data underlying the findings in their manuscript fully available?

Reviewer #1: Yes

Reviewer #2: Yes

4. Is the manuscript presented in an intelligible fashion and written in standard English?

Reviewer #1: No

Reviewer #2: Yes

5. Review Comments to the Author

Reviewer #1: This is a great opportunity to use twin designs to benefit the mental health of us all at this time. However, this manuscript requires some further editing to clarify the details of this study, as outlined in the feedback below. It also looks like you've done a lot with the twin data you have and you may like to be more selective about what you include. Some of the more minor comparisons could be left out and used in another context?

Abstract:

Line 20-21: "Participants in this study completed an online survey examining a number of health-related behaviors and outcomes and their impact due to COVID-19 mitigation". Are outcomes and impact synonymous?

Line 36-38: "These relationships are confounded by genetic and shared environmental factors, in the case of stress, and age and sex, in the case of anxiety." Be more specific when discussing confounding so as not to negate the value of your results entirely.

Also:

Add regression results (with effect sizes and p-values) to demonstrate the value in the paper beyond the descriptive statistics.

Name your measures and statistical designs in the Abstract.

Introduction:

Line 41: U.S. should be spelt out in full (first mention)

Line 43: Should read "... have been widely instituted..."

Line 62: Leading this section with "An important strength of the current study..." is more appropriate for arguing the study's strengths in the Discussion section.

Line 64: Explain clearly the role of twin data in your study/the twin designs. Point out your aims in using the twin designs. This is particularly important for those unfamiliar with twin designs who are reading your manuscript. Eg. i. ACE model to determine x, y, z. ii. Within- and between-twin design to determine x, y, z.

Also, are you able to discuss any existing findings from other studies to make the case for your study?

Materials and Methods:

Line 77: Add reference to questionnaire

Line 109: Should read: "Participants were asked about their feelings and thoughts in the last two weeks including the day the participant completed the survey..."?

Line 117: Should read "including the day the participant completed the survey"?

Line 125: "Considering the non-linear associations between perceived change in physical..." How do the figures show the non-linear associations?

Line 128: Should read: "Two comparisons were thus performed: decreased versus no change and increased versus no change; with each conducted separately for perceived stress and anxiety."

Line 131: Include the model numbers with each model in your methods to guide the reader

Line 141: Define b-subscript-A and b-subscript-C to make it clear what they represent. Eg. genetic (b-subscript-A a)

Line 150: "whereas twin analyses were performed only on the same-sex twins sample". Explain why you did this.

Results:

Explain the significance of between and within-pair comparisons

Use p-values with estimates when discussing evidence/significance of associations to help the reader easily understand your interpretations.

Discussion:

Line 219: Should read: "The phenotypic association between a decrease in physical activity and change in/increase in stress"?

Line 223: Should read: "the phenotypic association between a decrease in physical activity and change in/increase in anxiety"?

Line 249: U.S. (as used previously)

Line 251: "...it is likely ..." Do you have evidence for this? Otherwise you might like to rephrase this as "...possible reasons for a decrease in physical activity include..." (You mention in Line 255 "On the other hand, these measures may also have been

the reason why a small proportion of individuals reported an increase in physical activity."

Line 283: "On the other hand, social restriction measures varied greatly by country..." Is the discussion of country differences relevant to members of a U.S. based twin registry?

Line 285: Should read: "It was also difficult..." (Discussion has moved to a different topic)

Line 287: "At the time the survey was administered..." (Again, use "In addition" or other similar phrase to indicate discussion of new topic)

Also, are there any other studies you can compare and contrast to yours to in the Discussion?

Reviewer #2: Thank you for this opportunity to review your interesting and meaningful paper. The paper reports important and timely findings in an organised manner. This paper can further be strengthend by addressing the points below.

1] Abstract

key points were reported. Methods should include analyses. Results were a bit unclear, partly because above is unclear; can be clearer.

2] L40 - correct spell-out for covid-19 should be noted

3] Why did you only evaluate anxiety and stress? For example, another common mental health problem is depression. Justificaiton for inclusion of anxiety and stress needs to be discussed.

4] Sentences above "L73 Materials and Methods" are mixture of present and past tense, need to be consistent where appropriate. Also there are some errors e.g., "have" instead of "has". Proofreading is recommended.

5] L79 - you collected data from March 26. You need to note somewhere the start date of covid-19 associated mitigations. L271 reports it but it's in California.

6] L98, this survey is not relevant to recall bias, as your study looks at changes which include 'now'. This sentence should be removed. When you specifically talk about the past only, then this would apply.

7] What were the reliabilities of those scales you used?

8] Basic data of your sample were missing. what were the reliabilities of your data? outliers? normal distribution? Perceived stress and anxiety are square-root-transformed (L146), but reasons (e.g., shapiro-wilk) were not provided.

9] Age categorised by 10 is not well explained.

10] Figures need description

11] L245 - power analysis to be done to examine whether this statement stands or not

12] L272 - Why did you refer to California?

6. PLOS authors have the option to publish the peer review history of their article (what does this mean?). If published, this will include your full peer review and any attached files.

Reviewer #1: No

Reviewer #2: **Yes: **Yasuhiro Kotera

---

## [Author Response · Author response to Decision Letter 0]

29 Jul 2020

A detailed response to reviewers was included in the attached files.

---

## [Editor Report · Decision Letter 1]

3 Aug 2020

Perceived change in physical activity levels and mental health during COVID-19: Findings among adult twin pairs

PONE-D-20-20044R1

Dear Dr. Duncan,

We’re pleased to inform you that your manuscript has been judged scientifically suitable for publication and will be formally accepted for publication once it meets all outstanding technical requirements.

Kind regards,

Michio Murakami

Academic Editor

PLOS ONE
---

## [Editor Report · Acceptance letter]

5 Aug 2020

PONE-D-20-20044R1 

Perceived change in physical activity levels and mental health during COVID-19: Findings among adult twin pairs 

Dear Dr. Duncan:

I'm pleased to inform you that your manuscript has been deemed suitable for publication in PLOS ONE. Congratulations! Your manuscript is now with our production department. 

Kind regards, 

on behalf of

Dr. Michio Murakami 

Academic Editor

PLOS ONE